# Quality Assessment of *Artemisia rupestris* L. Using Quantitative Analysis of Multi-Components by Single Marker and Fingerprint Analysis

**DOI:** 10.3390/molecules27092634

**Published:** 2022-04-20

**Authors:** Xueqin Cao, Muchun Li, Liuchun Ma, Miaomiao Wang, Xueling Hou, Maitinuer Maiwulanjiang

**Affiliations:** 1State Key Laboratory Basis of Xinjiang Indigenous Medicinal Plants Resource Utilization, Xinjiang Technical Institute of Physics and Chemistry, Chinese Academy of Sciences, Urumqi 830011, China; caoxueqin8095@163.com (X.C.); limuchun20@mails.ucas.ac.cn (M.L.); wangmiao.1988.li@163.com (M.W.); xlhou@ms.xjb.ac.cn (X.H.); 2University of Chinese Academy of Sciences, Beijing 100049, China; 3Xinjiang Uyger Autonomous Region Academy of Instrumental Analysis, Urumqi 830011, China; 15199122582@163.com

**Keywords:** *Artemisia rupestris* L., high-performance liquid chromatography, quantitative analysis of multi-components by single marker, fingerprint analysis, quality control

## Abstract

The chromatographic fingerprint of 14 batches of *Artemisia rupestris* L. samples were established in this study. The constituents of ten components in *Artemisia rupestris* L. were determined using quantitative analysis of multi-components by single marker (QAMS) and the external standard method (ESM). Due to their stability and accessibility, chlorogenic acid and linarin were used as references to calculate the relative correction factors (RCFs) of apigenin-C-6,8-pentoside-hexoside, apigenin-C-6,8-di-pentoside, luteolin, 3,4-dicaffeoylquinic acid, 3,5-dicaffeoylquinic acid, 4,5-dicaffeoylquinic acid, chrysosplenetin B, and sbsinthin, based on high-performance liquid chromatography (HPLC). The value calculated by QAMS was consistent with that of the ESM, and the reproducibility of RCFs was found to be reliable. In conclusion, simultaneous determination of the ten components by the QAMS method and chromatographic fingerprint analysis were feasible and accurate in evaluating the quality of *Artemisia rupestris* L. and can be used as reference in traditional Chinese medicine quality control.

## 1. Introduction

*Artemisia rupestris* L. is a commonly used herbal medicine in Xinjiang for reducing fever and other symptoms of cold by acting as an anti-inflammatory and analgesic agent, as well as for detoxification and treating hepatitis [1]. Wild *A. rupestris* is distributed in Xinjiang, Middle Asian countries, and Northern Europe [2]. *A. rupestris* extracts contain sesquiterpenoids [3,4], flavonoids [5,6], alkaloids [7,8] and volatile oils [9,10]. 

A. rupestris has certain antiviral [11,12], anti-inflammatory activities and immune regulation [13,14,15] properties. Many studies have been conducted on the analysis of related components of A. rupestris and its preparations. In a previous study, the quality of Yizhihao capsules was assessed by quantitative analysis of rupestonic acid using the HPLC method [16]. Lan et al. [17] established an HPLC method to simultaneously determine rupestonic acidartesunone, chlorogenic acid, and luteolin in *A. rupestris*. Zhang Suwan et al. [18] determined the content of rupestonic acidartesunone, 6-dimethoxy-4, methyl artemisinin, and artemisinin in *A rupestris* using HPLC. Furthermore, Cai Xiaocui et al. [19] simultaneously determined the contents of chlorogenic and rupestonic acid, luteolin, vitexin, and apigenin in *A. rupestris* by liquid chromatography tandem-mass spectrometry (LC-MS/MS). 

In the global market, herbal medicines are treated or dried using different methods, which may result in unstable levels of single components. Fingerprinting can comprehensively reflect the overall chemical information of traditional Chinese medicine (TCM). It is usually used for origin identification, species certification and quality control of herbal medicine, and to evaluate the authenticity, excellence and stability of the quality of traditional Chinese medicine and semi-finished products of traditional Chinese medicine preparations [20,21]. By confirming the main common fingerprint peaks in the HPLC fingerprint, the qualitative and quantitative research of traditional Chinese medicine and its preparations can be carried out to evaluate and control quality [22,23].

The determination of the content of a single component cannot be used to accurately and sufficiently evaluate the quality of *A. rupestris*. Therefore, an effective strategy in determining the quality of *A. rupestris* will be to evaluate multiple components. In this study, we established a qualitative fingerprint method for *A. rupestris*, and ten principal components were confirmed and analyzed quantitatively.

QAMS enables the quantitative analysis of multiple components by using a cheap and easily available standard [24]. The combination of QAMS and fingerprint method showed the convenience and economic advantages of the QAMS method, meanwhile exhibiting the integrity and comprehensive advantages of the fingerprint method. Based on the chromatographic fingerprinting method and QAMS, the qualitative and quantitative determination of *A. rupestris* was evaluated in the present study. Chlorogenic acid and linarin were used as internal references for phenolic acids and flavonoids, respectively, to calculate the average RCFs. The proposed method of QAMS and fingerprint analysis provides a reliable, comprehensive and efficient way for evaluating *A. Rupestris* quality.

## 2. Results and Discussion

### 2.1. HPLC Conditions

Due to the complexity of chemical constituents in *A. rupestris*, it is crucial to separate the target components efficiently by optimizing chromatographic conditions.

The HPLC chromatographic peaks of *A. rupestris* were most informative with the ultraviolet (UV) wavelength of detection set at 350 nm. Therefore, we chose 350 nm for the assay of the selected components. The mobile phase consisted of acetonitrile-0.2% phosphoric acid at 1.0 mL/min of flow rate. The gradient elution program used is described in Section 3.5 and the favorable column temperature was set at 35 °C. The extraction solvent was ethanol-water (7:3, *v/v*) solution, and samples were treated for 30 min by ultrasonic extraction. The samples of *A. rupestris*, and the mixed standard solutions containing the 10 reference substances, were analyzed under the conditions described in Section 3.5. The chromatographic peak position of the standard substances was determined as follows: chlorogenic acid, apigenin-C-6,8-pentoside-hexoside, apigenin-C-6,8-di-pentoside, luteolin, 3,4-dicaffeoylquinic acid, 3,5-dicaffeoylquinic acid, 4,5-dicaffeoylquinic acid, linarin, chrysosplenetin B, and sbsinthin (Figure 1). 

According to the retention time in the chromatogram of the sample and standard solution, peaks 1, 2, 3, 4, 5, 6, 7, 8, 9, and 10 were identified to be chlorogenic acid, apigenin-C-6,8-pentoside-hexoside, apigenin-C-6,8-di-pentoside, luteolin, 3,4-dicaffeoylquinic acid, 3,5-dicaffeoylquinic acid, 4,5-dicaffeoylquinic acid, linarin, chrysosplenetin B, and sbsinthin, respectively. In addition, each peak was well separated in the present HPLC system. 

### 2.2. Method Validation

In order to support its application in the quantitative analysis of the ten compounds, the HPLC method was validated in terms of linearity, stability, precision, and accuracy.

#### 2.2.1. Linearity

Within the setting concentration range, the extent of direct linear relationship between the test results and the concentration of analytes in the samples was investigated. The mixed standard solutions were serially diluted to obtain desired concentrations with methanol. The 10 standard solutions were analyzed, and the calibration curves were formed by the relationship between the peak area of each component and corresponding concentration. The standard curve of each component was stable and the obtained linear regression equation was suitable for QAMS analysis (Table 1). 

#### 2.2.2. Stability

We investigated whether the sample solution was stable for 24 h storage at room temperature. The sample solution stability (S1) at room temperature (22 ± 3 °C) was tested at 0, 4, 8, 12, 16, 20, and 24 h, to obtain the RSDs. The RSD values of the stability tests were <2% (Table 1). The method could be considered stable, suggesting that within 24 h the sample solution was stable. 

#### 2.2.3. Accuracy

To verify the accuracy of the study, the mixed standard solutions of the analytes at low, medium, and high concentration levels were added into a sample (S1) of a certain amount (0.5 g), using six replicates. Then the mixed samples were subsequently extracted and analyzed. The average recovery of the 10 components was between 86.1–106%, and the RSDs of the accuracy values are shown in Table 2. 

#### 2.2.4. Precision

To evaluate the intra-day and inter-day precision, the sample solution was analyzed within a single assay day and on three separate days for six replicates at each concentration level, respectively. The RSDs of intra-day and inter-day were all less than 3% (Table 2). 

The HPLC method was verified in terms of stability, accuracy, and precision (Table 1 and Table 2), and the results suggested that this verified method was stable, accurate, precise and reproducible. Therefore, it could be used to determine chlorogenic acid, apigenin-C-6,8-pentoside-hexoside, apigenin-C-6,8-di-pentoside, luteolin, 3,4-dicaffeoylquinic acid, 3,5-dicaffeoylquinic acid, and 4,5-dicaffeoylquinic acid, linarin, chrysosplenetin B, and sbsinthin simultaneously in the *A. rupestris* samples. 

### 2.3. Fingerprints Analysis

Traditional Chinese medicine fingerprint is a comprehensive and quantifiable identification method, and is mainly used to evaluate the authenticity and the quality of traditional Chinese medicine and semi-finished products of traditional Chinese medicine preparations. For establishment of a novel method for multiple components from *A. rupestris*, high-performance liquid chromatograms of each sample were imported into the software recommended by SFDA, which was called chromatographic fingerprint similarity evaluation system for traditional Chinese medicine (version 2012a). Then, the chromatographic fingerprints were collected. Among the 14 samples, S1 was selected as the reference spectrum, while 18 common fingerprint peaks were recorded in the chromatograms (Figure 2). Meanwhile, the similarity of the 14 batches of *A. rupestris* samples was analyzed and obtained (Table 3). S11 gave the lowest similarity among the samples, and the simiarity of S1, S2, S3, S5, S6, S10, S14 were all above 0.8. According to the software requirements, similarity greater than 0.9 is generally required. In the similarity analysis, only 3 batches of samples were greater than 0.9, which were S1, S2 and S5, suggesting that the quality of medicinal materials from different producing areas and growth conditions is quite different.

### 2.4. Quantitative Analysis by QAMS and ESM

Ten components were identified in the fingerprint with reference materials, including chlorogenic acid, apigenin-C-6,8-pentoside-hexoside, apigenin-C-6,8-di-pentoside, luteolin, 3,4-dicaffeoylquinic acid, 3,5-dicaffeoylquinic acid, 4,5-dicaffeoylquinic acid, linarin, chrysosplenetin B, and sbsinthin. ESM and QAMS were used to quantify the ten components in the sample to verify consistency between QAMS and ESM. In ESM, the mixed standard solution and sample solutions were analyzed by liquid chromatography, and the content of each component calculated according to the regression equations listed in Table 1.

Choosing a suitable standard for internal reference using the QAMS method for analyzing multiple components in traditional Chinese medicinal substances is important. The component selected as internal reference should be selected on grounds of ease of acquisition, low price, and stable properties, and should be separable from the other compounds under chromatographic conditions [25]. In the present study, chlorogenic acid and linarin were used as internal references for phenolic acids and flavonoids. respectively. 

QAMS calculates the RCF between the component which was selected as internal reference and other components in medicinal materials. Furthermore, by calculating the amounts of other components through RCF, the simultaneous determination of multiple components [24,26] can be accomplished. The deviations (RE) between QAMS and ESM were calculated using the following formula (Equation (1)).
RCF = (C_i_/A_i_)/(C_s_/A_s_)(1)
where A_s_ is the peak area of the internal reference substance, C_s_ is the concentration of the internal reference substance, A_i_ is the peak area of the component to be tested, and C_i_ is the concentration of the component to be tested.

To analyze the 10 components simultaneously in *A. rupestris* by QAMS, the RCFs were calculated based on the ratio of peak area and corresponding concentration between internal references and the other analytes. The RCFs are shown in Table 4 and Table 5.

Overall, 14 batches of *A. rupestris* samples from various production areas were analyzed using the validated ESM and QAMS methods. The deviations (RE) between QAMS and ESM were calculated using the following formula (Equation (1)).
RE = (QAMS − ESM)/ESM × 100%(2)

The quantitative results of the 10 compounds in *A. rupestris* calculated by ESM and QAMS methods are shown in Table 6 and Table 7. The REs were less than 5%, which is the requirement of Chinese Pharmacopoeia. It was indicated that there was no significant difference in the content results obtained by QAMS and ESM

Analysis of the 14 batches of *A. rupestris* samples revealed the contents of chlorogenic acid, apigenin-C-6,8-pentoside-hexoside, apigenin-C-6,8-di-pentoside, luteolin, 3,4-dicaffeoylquinic acid, 3,5-dicaffeoylquinic acid, 4,5-dicaffeoylquinic acid, linarin, chrysosplenetin B, and sbsinthin. The determination acquired by QAMS was consistent with that of ESM. Therefore, it was proven that the QAMS method for simultaneous quantitative analysis of the 10 components in *A. rupestris* was reliable and feasible.

Fingerprints analysis and quantitative data showed significant differences among various samples from different sources. According to the results in Table 6 and Table 7, it can be seen that the content of ten components fluctuates greatly between batches. It may be related to the production environment, planting methods, growth time and harvest season. Among phenolic acids, the contents of chlorogenic acid and 3,5-dicaffeoylquinic acid were relatively high, and there was positive correlation. Among the flavonoids, the contents of linarin and chrysosplenetin B were relatively high. Next, we studied the relationship between the content of each component and the related efficacy.

### 2.5. Cluster Analysis

All the 14 samples of *A. rupestris* were selected from different regions or planting methods, and the content of their 10 main compounds might differ. Our QAMS methods could accurately analyze the composition, and the contents of the 10 compounds in the 14 batches of *A. rupestris* samples selected as clustering variable. The samples were divided into two categories by cluster analysis. S2, S3, and S4 are one category, which were artificially planted in Fuyun County of Altay Region and the other samples formed one category, to ascertain whether artificial planting or origins would change the composition of *A. rupestris*.

## 3. Materials and Methods

### 3.1. Plant Material

*A. rupestris* samples used in the study were collected from different production areas, as shown in Table 8.

### 3.2. Chemicals

Standard substances of chlorogenic acid (Batch No. 110753–201817, purity HPLC ≥ 96.8%), luteolin (Batch No. 111720–201810, purity HPLC ≥ 93.5%), 3,5-dicaffeoylquinic acid (Batch No. 111782–201807, purity HPLC ≥ 94.3%), 4,5-dicaffeoylquinic acid (Batch No. 111894–201102, purity HPLC ≥ 94.1%), linarin (Batch No. 111528–201911, purity HPLC ≥ 98.5%), and sbsinthin (Batch No. 111879–201102, purity HPLC ≥ 97.2%) were purchased from the National Institute for Food and Drug Control, (Beijing, China). 3,4-dicaffeoylquinic acid (Batch No. S0990020, purity HPLC ≥ 98.7%) was purchased from ANPEL Laboratory Technologies Inc., (Shanghai, China), chrysosplenetin B (Batch No. PRF10121942, purity HPLC ≥ 98.0%) was purchased from Biopurify Phytochemicals Ltd. (Chengdu, China). Apigenin-C-6,8-pentoside-hexoside and apigenin-C-6,8-di-pentoside were determined based on their spectral structure, and their purities were calculated by the peak area normalization method (purity HPLC ≥ 98.0%). Acetonitrile and methanol (Thermo Fisher Scientific, Inc. Shanghai, China) were of HPLC grade. Phosphoric acid (YSHC Chemical Company Limited, Tianjin, China) and absolute ethanol (Tianjin Xinbote Chemical Company Limited, Tianjin, China) were analytical grade. Ultrapure water was prepared by a Milli-QAC SP Reagent Water System (Millipore Corporation, Billerica, MA, USA). Other chemicals used in the study were all analytical grade.

### 3.3. Procedure of Sample Solution Preparation

The *A. rupestris* samples were ground using a high-speed traditional Chinese medicine pulverizer to make powder, and passed through a 10-mesh sieve. Next, 0.5 g of sample was weighed precisely into a bottle with a plug. The sample powder was extracted with 30 mL of the extraction solvent (ethanol:water, 7:3, v/v). The mixed solution was sonicated for 30 min (250 W, 40 kHz) at room temperature (25 ± 5 °C). After cooling, additional extraction solvent was added to the sample solution to compensate for weight loss, followed by thorough shaking. Before injection, the supernatant of the sample solution was filtered through membranes of 0.22 μm.

### 3.4. Preparation of the Reference Solution

Appropriate amounts of chlorogenic acid, apigenin-C-6,8-pentoside-hexoside, apigenin-C-6,8-di-pentoside, luteolin, 3,4-dicaffeoylquinic acid, 3,5-dicaffeoylquinic acid, 4,5-dicaffeoylquinic acid, linarin, chrysosplenetin B, and sbsinthin were weighed, and methanol were added to make 1.0 mg/mL stock solutions. These solutions were diluted with methanol serially, thereby making a mixture to obtain the desired concentrations for establishing the calibration curves of phenolic acids and flavonoids.

### 3.5. Instruments and Chromatographic Procedures

An Agilent 1260 series high-performance liquid chromatograph, equipped with a quaternary pump, UV detection, an autosampler, a column temperature controller, and a vacuum degasser (Agilent Technologies, Palo Alto, CA, USA) was used in the HPLC assay of *A. rupestris*. The samples were separated using a Hypersil GOLD C_18_ column (5 μm, 250 × 4.6 mm). The injection volume of sample and standards were all 10 μL, and the column temperature was maintained at 35 °C. A mixture of acetonitrile (A) and 0.2% phosphate solution (B) was used as the mobile phase at a flow rate of 1.0 mL/min. The gradient elution mode was modified as follows: 0–15 min, 5–18% A; 30 min, 20% A; 35 min, 21% A; 40 min, 40% A; 47 min, 45% A; 51–55 min, 80% A; 55.1–65.0 min, 5% A. The detection wavelength was sat at 350 nm. 

## 4. Conclusions

A novel method to determine multiple components from *A. rupestris* was established and reported using QAMS in this study, which was reproducible and accurate. The A. rupestris samples from different regions or planting methods were determined by HPLC, and the content determination of ten components (chlorogenic acid, apigenin-C-6,8-pentoside-hexoside, apigenin-C-6,8-di-pentoside, luteolin, 3,4-dicaffeoylquinic acid, 3,5-dicaffeoylquinic acid, 4,5-dicaffeoylquinic acid, linarin, chrysosplenetin B, and sbsinthin in different laboratories) in the *A. rupestris* samples was simultaneously accomplished by the proposed QAMS, applying an exclusive identification and evaluation method for qualitative and quantitative analysis of *A. rupestris.*

The chromatographic fingerprint showed the details of *A. rupestris* chromatographic spectrum, while the cluster analysis identified that region and growth conditions could influence the content of *A. rupestris*. Therefore, this method might be suitable both for A. rupestris quantitative analysis and for quality examination. The similarity analysis suggested that the quality of medicinal materials from different producing areas and growth conditions is quite different.

Thus, the proposed method could be an accurate and feasible approach for provision of supporting qualitative and quantitative data for *A. rupestris* quality evaluation. QAMS can be applied to determine chlorogenic acid, apigenin-C-6,8-pentoside-hexoside, apigenin-C-6,8-di-pentoside, luteolin, 3,4-dicaffeoylquinic acid, 3,5-dicaffeoylquinic acid, 4,5-dicaffeoylquinic acid, linarin, chrysosplenetin B, and sbsinthin simultaneously, and there might be potential for establishment of a universal and unified standard for the quality control of *A. rupestris*.

## Figures and Tables

**Figure 1 molecules-27-02634-f001:**
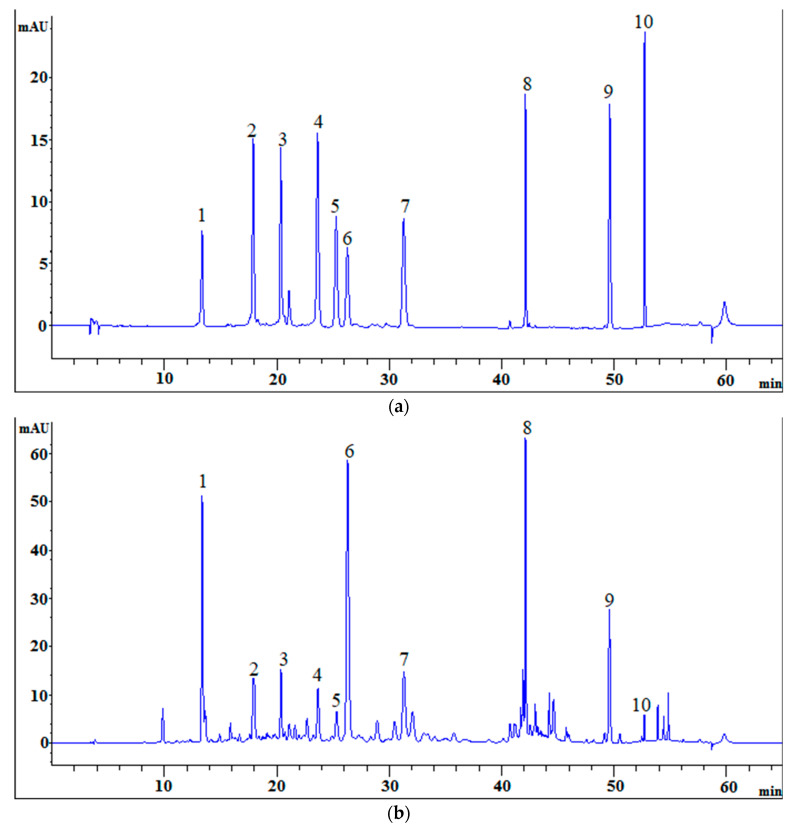
HPLC chromatograms of the mixed standards and *A. rupestris* sample. (**a**) the mixed standards; (**b**) *A. rupestris* sample. 1—Chlorogenic acid; 2—Apigenin-C-6,8-pentoside-hexoside; 3—Apigenin-C-6,8-di-pentoside; 4—Luteolin; 5—3,4-Dicaffeoylquinic acid; 6—3,5-Dicaffeoylquinic acid; 7—4,5-Dicaffeoylquinic acid; 8—Linarin; 9—Chrysosplenetin B; 10—Sbsinthin.

**Figure 2 molecules-27-02634-f002:**
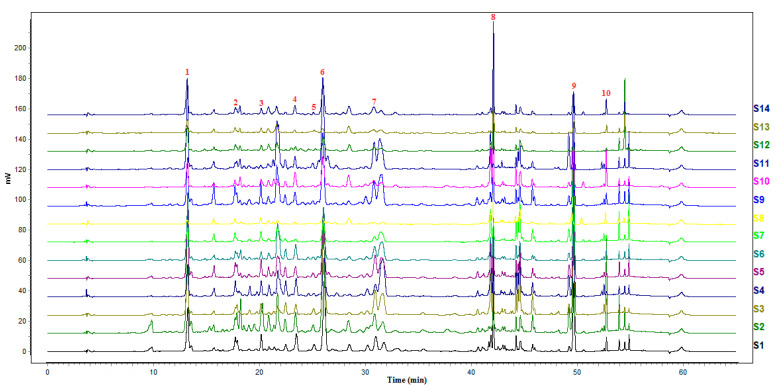
The fingerprints of *A. rupestris* samples. 1—Chlorogenic acid; 2—Apigenin-C-6,8-pentoside-hexoside; 3—Apigenin-C-6,8-di-pentoside; 4—Luteolin; 5—3,4-Dicaffeoylquinic acid; 6—3,5-Dicaffeoylquinic acid; 7—4,5-Dicaffeoylquinic acid; 8—Linarin; 9—Chrysosplenetin B; 10—Sbsinthin.

**Table 1 molecules-27-02634-t001:** The Regression Equation, Linear Range, and Stability of the 10 Components Analyzed by HPLC (*n* = 6).

Analytes	Regression Equations	r^2^	Linear Ranges (μg/mL)	Stability (RSD%)
Chlorogenic acid	Y = 12.8332 X + 0.3077	0.999 4	0.50–100	0.4
Apigenin-C-6,8-pentoside-hexoside	Y = 19.858 X − 1.2861	0.999 5	0.50–50.0	0.6
Apigenin-C-6,8-di-pentoside	Y = 20.056 X + 1.8584	0.999 3	0.50–50.0	0.8
Luteolin	Y = 26.19249 X + 1.2730	0.999 0	0.50–50.0	0.9
3,4-Dicaffeoylquinic acid	Y = 15.76744 X + 1.36516	0.999 3	0.50–100	0.5
3,5-Dicaffeoylquinic acid	Y = 13.62685 X + 1.25882	0.999 2	0.50–100	0.7
4,5-Dicaffeoylquinic acid	Y = 19.40530 X + 2.40872	0.999 1	0.50–100	1.1
Linarin	Y = 18.4154 X + 1.7983	0.999 8	0.50–50.0	0.5
Chrysosplenetin B	Y = 36.9728 X + 2.5677	0.999 8	0.50–50.0	0.9
Sbsinthin	Y = 18.5140 X + 1.4366	0.999 9	0.50–50.0	1.2

Y—peak area; X—concentration (μg/mL); r^2^—correlation coefficient of the equation.

**Table 2 molecules-27-02634-t002:** Accuracy and Precision of the 10 Components.

Analytes	Recovery (*n* = 6)						Precision (*n* = 6)	
	Added Level	Recovery (%)	Added Level	Recovery (%)	Added Level	Recovery (%)	Intra-Day ^a^ (%RSD)	Inter-Day ^b^ (%RSD)
Chlorogenic acid	Low	89.6	Middle	90.2	High	93.1	1.9	2.5
apigenin-C-6,8-pentoside-hexoside	Low	88.3	Middle	93.6	High	104	2.2	2.4
Apigenin-6, 8-di-C-pen	Low	90.1	Middle	101	High	96.1	1.8	2.1
Luteolin	Low	93.1	Middle	98.1	High	98.2	2.0	2.4
3,4-Dicaffeoylquinic acid	Low	89.5	Middle	102	High	103	1.8	2.5
3,5-Dicaffeoylquinic acid	Low	86.1	Middle	96.3	High	101	1.3	1.6
4,5-Dicaffeoylquinic acid	Low	90.2	Middle	95.2	High	95.4	1.4	1.5
Linarin	Low	103	Middle	106	High	96.1	2.1	2.4
Chrysosplenetin B	Low	91.5	Middle	91.4	High	92.4	0.9	2.6
Sbsinthin	Low	92.8	Middle	95.3	High	99.8	1.8	1.9

^a^ Intra-day precision tested six times during the same day. ^b^ Inter-day precision tested on three separate days.

**Table 3 molecules-27-02634-t003:** The Similarity of 14 batches *A. rupestris* Samples.

No.	Similarity	No.	Similarity
S1	0.949	S8	0.766
S2	0.916	S9	0.672
S3	0.858	S10	0.811
S4	0.744	S11	0.620
S5	0.917	S12	0.693
S6	0.824	S13	0.712
S7	0.770	S14	0.897

**Table 4 molecules-27-02634-t004:** Relative Correction Factor (RCF, *f*_x_) Values of Phenolic Acids.

Con. (μg/mL)	*f* _chlorogenic acid/3,4-Dicaffeoylquinic acid_	*f* _chlorogenic acid/_ _3,5-Dicaffeoylquinic acid_	*f* _chlorogenic acid/_ _4,5-Dicaffeoylquinic acid_
0.50	0.918	0.954	0.713
2.0	0.877	1.024	0.709
5.0	0.871	1.015	0.697
10.0	0.841	0.980	0.661
50.0	0.837	0.962	0.685
100	0.843	0.990	0.684
Means	0.865	0.987	0.692
RSD (%)	3.6	2.8	2.8

**Table 5 molecules-27-02634-t005:** Relative Correction Factor (RCF, *f*_x_) Values of Flavonoids.

Con. (μg/mL)	*f* _linarin/_ _apigenin-C-6,8-pentoside-hexoside_	*f* _linarin/_ _apigenin-C-6,8-di-pentoside_	*f* _linarin/luteolin_	*f* _linarin/chrysosplenetin B_	*f* _linarin/sbsinthin_
0.50	0.663	0.492	0.847	0.958	0.885
1.0	0.707	0.504	0.870	0.951	0.879
2.0	0.707	0.505	0.876	0.943	0.871
5.0	0.727	0.511	0.893	0.951	0.879
10.0	0.709	0.499	0.876	0.949	0.876
20.0	0.684	0.482	0.844	0.902	0.833
Means	0.700	0.499	0.868	0.942	0.870
RSD (%)	3.2	2.1	2.1	2.2	2.2

**Table 6 molecules-27-02634-t006:** Contents of Phenolic Acids in *A. rupestris* using different methods (mg/g).

No.	Chlorogenic Acid	3,4-Dicaffeoylquinic Acid			3,5-Dicaffeoylquinic Acid			4,5-Dicaffeoylquinic Acid		
	ESM	ESM	QAMS	RE/%	ESM	QAMS	RE/%	ESM	QAMS	RE/%
S1	1.685	0.266	0.278	4.3	3.326	3.210	3.5	0.574	0.586	2.1
S2	5.223	0.279	0.291	4.5	5.506	5.310	3.6	0.746	0.761	2.0
S3	2.013	0.238	0.249	4.6	4.789	4.612	3.7	0.949	0.964	1.6
S4	1.247	0.096	0.099	3.2	1.934	1.874	3.1	0.402	0.412	2.5
S5	2.708	0.150	0.156	4.1	3.512	3.392	3.4	0.885	0.901	1.8
S6	2.528	0.110	0.112	1.5	2.457	2.381	3.1	0.552	0.564	2.3
S7	0.559	0.0580	0.0590	1.8	0.936	0.915	2.3	0.187	0.195	3.9
S8	0.132	0.0186	0.0190	3.4	0.294	0.294	0.1	0.0176	0.0180	2.3
S9	2.103	0.134	0.142	6.4	3.540	3.416	3.5	0.946	0.962	1.6
S10	3.163	N.D.	N.D.	/	1.560	1.515	2.9	0.391	0.400	2.4
S11	0.874	0.212	0.223	5.0	4.376	4.222	3.5	0.989	1.006	1.7
S12	1.008	N.D.	N.D.	/	0.699	0.693	0.8	0.168	0.176	5.0
S13	0.590	N.D.	N.D.	/	0.179	0.186	4.2	0.182	0.190	4.1
S14	1.470	N.D.	N.D.	/	1.580	1.538	2.7	0.359	0.369	2.8

N.D.: not detected.

**Table 7 molecules-27-02634-t007:** Contents of Flavonoids in *A. rupestris* using different methods (mg/g).

No.	Linarin	Apigenin-C-6,8-Pentoside-Hexoside			Apigenin-C-6,8-di-Pentoside			Luteolin			Chrysosplenetin B			Sbsinthin		
	ESM	ESM	QAMS	RE/%	ESM	QAMS	RE/%	ESM	QAMS	RE/%	ESM	QAMS	RE/%	ESM	QAMS	RE/%
S1	0.828	0.500	0.482	3.6	0.297	0.301	1.3	0.366	0.356	2.6	0.727	0.714	1.8	0.130	0.125	3.9
S2	1.298	0.521	0.514	1.3	0.898	0.905	0.8	0.392	0.379	3.3	1.060	1.080	1.9	0.859	0.852	0.9
S3	0.338	0.374	0.385	2.9	0.212	0.207	2.4	0.234	0.225	3.9	1.795	1.751	2.4	0.138	0.137	1.0
S4	0.409	0.390	0.393	0.8	0.286	0.281	1.7	0.354	0.344	2.8	1.346	1.339	0.5	0.0366	0.0362	1.0
S5	0.483	0.527	0.521	1.1	0.345	0.341	1.2	0.236	0.237	0.4	0.842	0.874	3.7	0.0401	0.0395	1.5
S6	0.428	0.279	0.285	2.2	0.174	0.176	1.1	0.324	0.326	0.7	0.509	0.500	1.7	0.0909	0.0889	2.2
S7	0.248	0.241	0.249	3.3	0.211	0.209	0.9	0.0522	0.0501	4.0	1.145	1.147	0.2	0.0532	0.0512	3.8
S8	0.558	N.D.	N.D.	/	0.596	0.600	0.7	0.0347	0.0344	0.7	0.124	0.126	1.3	0.103	0.107	3.4
S9	0.940	0.627	0.63	0.5	0.427	0.431	0.9	0.238	0.230	3.2	1.122	1.137	1.3	0.101	0.105	4.3
S10	1.090	0.151	0.153	1.3	0.180	0.177	1.7	0.270	0.261	3.4	0.552	0.543	1.7	0.408	0.398	2.4
S11	0.047	0.205	0.211	2.9	0.103	0.102	1.0	0.290	0.251	3.3	0.614	0.622	1.3	0.0550	0.0523	4.8
S12	0.645	0.158	0.155	1.9	0.107	0.105	1.9	0.0793	0.0753	5.0	0.0575	0.0549	4.5	0.0284	0.0288	1.3
S13	0.696	0.121	0.118	2.5	0.0953	0.0950	0.3	0.0512	0.0496	3.1	0.156	0.151	3.4	0.0783	0.0748	4.5
S14	1.076	0.188	0.190	1.1	0.119	0.121	1.7	0.174	0.171	1.6	0.280	0.267	4.6	0.150	0.146	2.3

**Table 8 molecules-27-02634-t008:** The different production areas of *A. rupestris*.

No.	Production Areas
S1	Urumqi, Xinjiang
S2	Altay Prefecture, Xinjiang
S3	Altay Prefecture, Xinjiang
S4	Altay Prefecture, Xinjiang
S5	Altay Prefecture, Xinjiang
S6	Urumqi, Xinjiang
S7	Urumqi, Xinjiang
S8	Urumqi, Xinjiang
S9	Fukang, Xinjiang
S10	Urumqi, Xinjiang
S11	Altay Prefecture, Xinjiang
S12	Urumqi, Xinjiang
S13	Urumqi, Xinjiang
S14	Fukang, Xinjiang

## Data Availability

Data are contained within the article.

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
