# Peer review of "Quality Assessment of Artemisia rupestris L. Using Quantitative Analysis of Multi-Components by Single Marker and Fingerprint Analysis"

_molecules, 2022, doi:10.3390/molecules27092634_

Round 1

Reviewer 1 Report

Dear Editor,

Overall, this article can be considered for publication in Molecules: An International Journal, in terms of data updates to methods for evaluating the quality of herbal medicines. Besides being able to provide new information about the components of a herbal medicine, it can also evaluate the quality of a plant used as a herbal medicine to become a reference for controlling the quality of herbal medicines in China. This can also be a consideration for researchers in developing new drugs. Overall, this paper is of great interest in drug development. However, there are some things that need to be emphasized and improved, such as the following suggestions:

  1. Abstract (Page 1 Line 13): Please don't use conjunctions (and) in front of sentences
  2. Abstract (Page 1, line 18-19): Authors must write the scientific reasoning why chlorogenic acid and and linarin as relative correction factors.
  3. The author must pay attention to the flow of article content thus the phrases or paragraphs are related so that the reader does not get confused reading it.

I found that many phrases do not have connecting sentences between paragraphs. For example, paragraph 1 (introduction) told about the plant Artemisia rupestris then the author suddenly wrote about the quality control of Yizhihao capsules. Readers will wonder what is this capsule? Or in paragraph 3, the authors told the analyst of the active component in Artemisia rupestris in the global market. However, suddenly the author described QAMS and RCF even authors stated this methods as novelty. It’s so confused.

  1. Authors mut write briefly about QAMS and RCF to make it easier for readers to understand this article.
  2. Introduction (Page 2): To bring up a novelty, it is better to search the literature related to previous research with the similarity of the sample and or the similarity of the possible components of the compound contained in the sample used.
  3. Introduction (Page 2, Line 50): The component fingerprinting method was mentioned in Artemisia rupestris L. but it was not explained for what purpose, considering that a quantitative analysis of the 10 main components in the sample using HPLC was also carried out.
  4. Results (Page 2 Line 70): I'm curious about Figure 1, is it correct to use HPLC and LC-MS or GS-MS?. Because the peak of the chromatogram is very sharp. Can the authors provide the original results from HPLC?
  5. Results and Discussion (Figure 1 and 2, line 68 and 81): there is no data related to the retention time of each compound and the difference between all samples used.
  6. Results and discussion (Figure 1 and 2, line 68 and 81): it is not explained which compound is attached at what retention time and what its use is so that this research is considered important.
  7. Results and Discussion (Page 3, line 94): It is not explained the meaning in the acquisition of values ​​for several parameters of the validation method used.
  8. Result and Discussion (Page 5, line 130): Not explained the good similarity acceptance value.
  9. Result and Discussion (Page 5, line 159): Authors must write down the reason for doing Quantitative Analysis using QAMS and ESM.
  10. Results and Discussion: There is no elaboration on the results of the statistical analysis.
  11. Since this manuscript has many abbreviations, it would be better if a list of abbreviations was added.
  12. Conclusion: If the author states that the QAMS study is a novelty in analyzing the components of this plant, the author should explain in more detail what QAMS is, if there is a mathematical equation from QAMS it should be written in the manuscript.

Author Response

Response to Reviewer 1 Comments

We thank the reviewer for the very helpful comments. We have studied reviewers’ comment very carefully and revised the manuscript according to the recommendations. Follows are our answers to the referee’s comments.

Point 1: Abstract (Page 1 Line 13): Please don't use conjunctions (and) in front of sentences.

Response 1: Thank you for your correction, we had corrected this mistake. (Page 1, Line 13)

Point 2: Abstract (Page 1, line 18-19): Authors must write the scientific reasoning why chlorogenic acid and and linarin as relative correction factors.

Response 2: According to your suggestion, the content of the manuscript had been supplemented. (Page 1, Line 15)

Point 3: The author must pay attention to the flow of article content thus the phrases or paragraphs are related so that the reader does not get confused reading it.

I found that many phrases do not have connecting sentences between paragraphs. For example, paragraph 1 (introduction) told about the plant Artemisia rupestris then the author suddenly wrote about the quality control of Yizhihao capsules. Readers will wonder what is this capsule? Or in paragraph 3, the authors told the analyst of the active component in Artemisia rupestris in the global market. However, suddenly the author described QAMS and RCF even authors stated this methods as novelty. It’s so confused.

Response 3: According to your suggestion, we have added relevant references (reference11-15, 20, 21). And the content of the manuscript had been supplemented and adjusted. (Page 1, Line 33-35; Page 2, Line 45-49)

  1. rupestris has certain antiviral [11, 12], anti-inflammatory activities and immune regulation [13-15]. Many studies have been carried on the analysis of related components of A. rupestris and its preparations.

Point 4: Authors mut write briefly about QAMS and RCF to make it easier for readers to understand this article.

Response 4: According to your suggestion, the content of the manuscript had been supplemented and adjusted. (Page 6, Line 203-207)

QAMS calculates the RCF between the component which was selected as internal references and other components in medicinal materials. Furthermore, by calculating the amounts of other components through RCF, the simultaneous determination of multiple components [21, 22] can be accomplished.

Point 5: Introduction (Page 2): To bring up a novelty, it is better to search the literature related to previous research with the similarity of the sample and or the similarity of the possible components of the compound contained in the sample used.

Response 5: According to your suggestion, the content of the manuscript had been supplemented. (reference 20, 21); (Page 2, Line 45-49)

Fingerprint can comprehensively reflect the overall chemical information of traditional Chinese medicine (TCM). By confirming the main common fingerprint peaks in the HPLC fingerprint, the qualitative and quantitative research of traditional Chinese medicine and its preparations can be carried out to evaluate and control the quality [20, 21].

Point 6: Introduction (Page 2, Line 50): The component fingerprinting method was mentioned in Artemisia rupestris L. but it was not explained for what purpose, considering that a quantitative analysis of the 10 main components in the sample using HPLC was also carried out.

Response 6: According to your suggestion, the content of the manuscript had been supplemented and adjusted. (Page 2, Line 52-61)

In this study, we established a qualitative fingerprint method for A. rupestris, and 10 principal components were confirmed and analyzed quantitatively.

Point 7: Results (Page 2 Line 70): I'm curious about Figure 1, is it correct to use HPLC and LC-MS or GS-MS?. Because the peak of the chromatogram is very sharp. Can the authors provide the original results from HPLC?

Response 7: The chromatograms of Figure 1were original results from HPLC, and we have corrected this mistake (a:the mixed standards A. rupestris sample; b: the mixed standardsA. rupestris sample) (Page 3, Line 107-109)

Point 8: Results and Discussion (Figure 1 and 2, line 68 and 81): there is no data related to the retention time of each compound and the difference between all samples used.

Response 8: We have marked the chromatographic peak on Figure 2. (Page 6, Line 180)

The chromatographic peak peaks 1, 2, 3, 4, 5, 6, 7, 8, 9, and 10 were identified to be chlorogenic acid, apigenin-C-6,8-pentoside-hexoside, apigenin-C-6,8-di-pentoside, luteolin, 3,4-dicaffeoylquinic acid, 3,5-dicaffeoylquinic acid, 4,5-dicaffeoylquinic acid, linarin, chrysosplenetin B, and sbsinthin.

Point 9: Results and discussion (Figure 1 and 2, line 68 and 81): it is not explained which compound is attached at what retention time and what its use is so that this research is considered important.

Response 9: According to your suggestion, the content of the manuscript had been supplemented and adjusted. (Page 2, Line 79-82)

Point 10: Results and Discussion (Page 3, line 94): It is not explained the meaning in the acquisition of values ​​for several parameters of the validation method used.

Response 10: According to your suggestion, the content of the manuscript had been supplemented. (Page 4, Line 120-121)

Point 11: Result and Discussion (Page 5, line 130): Not explained the good similarity acceptance value.

Response 11: According to your suggestion, the content of the manuscript had been supplemented and adjusted. (Page 5, Line 171-174)

According to the software requirements, it is generally requiring the similarity is greater than 0.9.

Point 12: Result and Discussion (Page 5, line 159): Authors must write down the reason for doing Quantitative Analysis using QAMS and ESM.

Response 12: According to your suggestion, the content of the manuscript had been supplemented. (Page 6, Line 192-196)

Ten components were identified in the fingerprint with reference materials. ESM and QAMS were used to quantify ten components in the sample to verify the consistency be-tween QAMS and ESM.

Point 13: Results and Discussion: There is no elaboration on the results of the statistical analysis.

Response 13: According to your suggestion, the content of the manuscript had been supplemented. (Page 9, Line 236-243)

Among phenolic acids, the contents of chlorogenic acid and 3,5-dicaffeoylquinic acid were relatively high, and there was positive correlation. Among the flavonoids, the contents of linarin and chrysosplenetin B were relatively high. Next, we will study the relationship between the content of each component and the related efficacy.

Point 14: Since this manuscript has many abbreviations, it would be better if a list of abbreviations was added.

Response 14: According to your suggestion, a list of abbreviations was added. (Page 11, Line 338-345)

Point 15: Conclusion: If the author states that the QAMS study is a novelty in analyzing the components of this plant, the author should explain in more detail what QAMS is, if there is a mathematical equation from QAMS it should be written in the manuscript.

Response 15: According to your suggestion, an equation was added and the content of the manuscript had been supplemented and adjusted. (Page 6, Line 208-211).

QAMS calculates the RCF between the component which was selected as internal references and other components in medicinal materials. Furthermore, by calculating the amounts of other components through RCF, the simultaneous determination of multiple components [21, 22] can be accomplished.

I would like to express my sincere thanks to reviewer 1 for the recognition of the article. Thank you very much.

We thank the reviewer and remain at your disposal for any further question.

Your sincerely,

Dr. Maitinuer Maiwulanjiang

Xinjiang Key Laboratory of Plant Resources and Natural Products Chemistry,

Xinjiang Technical Institute of Physics and Chemistry,

Chinese Academy of Sciences,

Beijing South Road 40-1

Urumqi, Xinjiang, 830011, China

Phone: (+86) 0991-6631740

Fax: (+86) 0991-3838957

Email: mavlanjan@ms.xjb.ac.cn

Reviewer 2 Report

The article sent for consideration for publication in a scientific journal by prof. Maiwulanjiang and colleagues concerns the development of a highly desirable method of determining the quality of a title herbal product using the fingerprint method. The method of marking is described in a very detailed manner, but it has all the features of a report and is out of date due to the lack of current references.

This article entitled “Quality Assessment of Artemisia rupestris L. using Quantitative Analysis of Multi-components by Single Marker and Fingerprint Analysis” is suitable for publication in the special issue of scientific journal Molecules after updating the state of knowledge, and discussion of more literature on research topics, in closely related fields, for example, the current state of knowledge about the properties and uses of such well-chosen reference substances and / or the state of the art in fingerprint analysis of related herbal medicines.

A few minor shortcomings to be corrected in this well-edited text are listed below:

At lines 65 and 226 authors used word “ ethanol “. Please add specification (at section 3.2.) such as absolute ethanol, spirit, rectified spirit, etc, and it customer.

At line 83 is: … a: A. rupestris … , but should be better: … a: A. rupestris … . Similar errors are at lines 195 and 256.

Comment: The Latin name should be italicized, see for example line 83.

At line 142 is: … and the smililarity … , but should be: … and the simililarity … .

Comment: please insert the missing letter “ i “.

At line 206 is: … ( Batch … , but should be: … (Batch … .

Comment: remove any unnecessary space.

At line 217 is: … methnol … , but maybe should be: … methanol … .

At line 293 is: … 2012 … , but should be: … 2012 … . Similar error to correct is in line 320.

Comment: The year of publication should be written in bold.

At lines 307–308 is unformal text: … (I was unable to source this article online in English. Please provide the names of all authors as this is required by the journal.) … .

Comment: Please delete the informal sentence and provide the correct surnames and initials of all authors' names in ref. 12, before the next stage of production.

Author Response

We are submitting our revised manuscript for your reconsideration of its suitability for publication. All authors have read and approved the manuscript. We have carefully taken the reviewers’ comments into account and provided responses to each of the points raised by the reviewers. Some necessary corrections have been made, and all the altered passages have been using the “Track Changes” function. Please refer to the present manuscript text.

We thank the reviewer for the very helpful comments. We have studied reviewers’ comment very carefully and revised the manuscript according to the recommendations. Follows are our answers to the referee’s comments.

Point 1: The method of marking is described in a very detailed manner, but it has all the features of a report and is out of date due to the lack of current references.

Response 1: According to your suggestion, we have added relevant references. (reference11-15, 20, 21), (Page 1, Line 33-35; Page 2, Line 45-49)

Point 2: At lines 65 and 226 authors used word “ ethanol “. Please add specification (at section 3.2.) such as absolute ethanol, spirit, rectified spirit, etc, and it customer.

Response 2: We had add specification and cutomer of ethanol at section 3.2. (Line 272-273)

Point 2: At line 83 is: … a: A. rupestris … , but should be better: … a: A. rupestris … . Similar errors are at lines 195 and 256.

Comment: The Latin name should be italicized, see for example line 83.

Response 2: Thank you for your correction, we had corrected this mistake. (Line 248, 310)

Point 4: At line 142 is: … and the smililarity … , but should be: … and the simililarity … .

Comment: please insert the missing letter “ i “.

Response 4: Thank you for your correction, we had corrected this mistake. (Line 169)

Point 5: At line 206 is: … ( Batch … , but should be: … (Batch … .

Comment: remove any unnecessary space.

Response 5: Thank you for your correction, we had corrected this mistake. (Line 259)

Point 6: At line 217 is: … methnol … , but maybe should be: … methanol … .

Response 6: Thank you for your correction, we had corrected this mistake. (Line 270)

Point 7: At line 293 is: … 2012 … , but should be: … 2012 … . Similar error to correct is in line 320.

Comment: The year of publication should be written in bold.

Response 7: Thank you for your correction, we had corrected this mistake. (Line 357 and 394)

Point 8: At lines 307–308 is unformal text: … (I was unable to source this article online in English. Please provide the names of all authors as this is required by the journal.) … .

Comment: Please delete the informal sentence and provide the correct surnames and initials of all authors' names in ref. 12, before the next stage of production.

Response 8: Thank you for your correction, we had corrected this mistake. (Line 381 - 382)

I would like to express my sincere thanks to reviewer 2 for the recognition of the article. Thank you very much.

We thank the reviewer and remain at your disposal for any further question.

Your sincerely,

Dr. Maitinuer Maiwulanjiang

Xinjiang Key Laboratory of Plant Resources and Natural Products Chemistry,

Xinjiang Technical Institute of Physics and Chemistry,

Chinese Academy of Sciences,

Beijing South Road 40-1

Urumqi, Xinjiang, 830011, China

Phone: (+86) 0991-6631740

Fax: (+86) 0991-3838957

Email: mavlanjan@ms.xjb.ac.cn

Reviewer 3 Report

This study describes a method for the assessment of the authenticity and quality of extracts from Artemisia rupretris. The study is well-designed and the conclusions are supported by the data.  Aside from some minor grammatical errors, listed below, the only revision that I thinks need to be made is in the description of the software used to analyze the chromatograms.  It is unclear whether this is a commercial software package that evaluates the similarity of chromatograms or if it is a program developed by the authors.  I believe that a better description of the program beyond stating that the chromatograms were imported into the software.

Minor revisions

line 120 change evaluated to evaluate

line 132 change authenticity of to authenticity and

line 138 change fingerprint to fingerprints

line 227 change ultrasonicated to sonicated

line 252 change accuracy to accurate

line 307 check note in reference 12

Author Response

We thank the reviewer for the very helpful comments. We have studied reviewers’ comment very carefully and revised the manuscript according to the recommendations. Follows are our answers to the referee’s comments.

Point 1: the only revision that I thinks need to be made is in the description of the software used to analyze the chromatograms. It is unclear whether this is a commercial software package that evaluates the similarity of chromatograms or if it is a program developed by the authors.  I believe that a better description of the program beyond stating that the chromatograms were imported into the software.

Response 1: According to your suggestion, the content of the manuscript had been supplemented. (Page 5, Line 163)

Point 2: I line 120 change evaluated to evaluate.

Response 2: Thank you for your correction, we had corrected this mistake. (Page 5, Line 147)

Point 3: line 132 change authenticity of to authenticity and.

Response 3: Thank you for your correction, we had corrected this mistake. (Page 5, Line 159)

Point 4: line 138 change fingerprint to fingerprints.

Response 4: Thank you for your correction, we had corrected this mistake. (Page 5, Line 165)

Point 5: line 227 change ultrasonicated to sonicated.

Response 5: Thank you for your correction, we had corrected this mistake. (Page 10, Line 281)

Point 6: line 252 change accuracy to accurate.

Response 6: Thank you for your correction, we had corrected this mistake. (Page 10, Line 305)

Point 7: line 307 check note in reference 12.

Response 7: Thank you for your correction, we had corrected this mistake. (Page 12, Line 380 - 382)

I would like to express my sincere thanks to reviewer 3 for the recognition of the article. Thank you very much.

We thank the reviewer and remain at your disposal for any further question.

Your sincerely,

Dr. Maitinuer Maiwulanjiang

Xinjiang Key Laboratory of Plant Resources and Natural Products Chemistry,

Xinjiang Technical Institute of Physics and Chemistry,

Chinese Academy of Sciences,

Beijing South Road 40-1

Urumqi, Xinjiang, 830011, China

Phone: (+86) 0991-6631740

Fax: (+86) 0991-3838957

Email: mavlanjan@ms.xjb.ac.cn

Round 2

Reviewer 1 Report

Dear Editor

I appreciate the efforts of the author to fix this. Articles can be considered for publication if they have been corrected according to the suggestions below:

Minor:

  1. Introduction (Page 2, Line 45): said “Fingerprint can comprehensively reflect the overall chemical information of traditional Chinese medicine (TCM). It will be better if it is explained what the sample of herbal medicine that use in TCM relate to previous research?
  2. Introduction (Page 2, Line 59): said “The combination of QAMS and fingerprint method showed the convenience and 59 economic advantages by QAMs method “ can explain why it has economic advantages from QAMS methode?
  3. Result (page 4 : Line 122) it will be better if you write the aims of linearity.
  4. Result (page 4 : Line 132) it will be better if you write the aims of stability.
  5. Result (page 5 : Line 172) According to the software requirements, it is generally requiring the similarity is greater than 0.9. In the similarity analysis, 3 batches are greater than 0.9, suggest ing that the quality of medicinal materials from different producing areas and growth conditions is quite different. Which sampel number of the 3 batches are greater then 0,9.
  6. Result (page 6 : Line 192) Ten components were identified in the fingerprint with reference materials. You can describe what they are.
  7. Result (page 7 : Line 222) The quantitative results of the 10 compounds in A. rupestris calculated by ESM and QAMS methods are shown in Tables 6 and 7, and the REs were less than 5%. does the RE requirement have to be 5%? If yes, please explain the reference of this requirement. And all the RE% results are under 5%, what it does mean?

Author Response

Dear reviewer:

We thank the reviewer for the very helpful comments. We have studied reviewers’ comment very carefully and revised the manuscript according to the recommendations. Follows are our answers to the referee’s comments.

Point 1: Introduction (Page 2, Line 45): said “Fingerprint can comprehensively reflect the overall chemical information of traditional Chinese medicine (TCM). It will be better if it is explained what the sample of herbal medicine that use in TCM relate to previous research?

 Response 1: According to your suggestion, the content of the manuscript had been supplemented. (Page 2, Line 46-50; reference 20, 21)

It is usually used for origin identification, species certification and quality control of herbal medicine, and to evaluate the authenticity, excellence and stability of the quality of traditional Chinese medicine and semi-finished products of traditional Chinese medicine preparations [20, 21].

Point 2: Introduction (Page 2, Line 59): said “The combination of QAMS and fingerprint method showed the convenience and economic advantages by QAMs method “ can explain why it has economic advantages from QAMS methode?

Response 2: According to your suggestion, the content of the manuscript had been supplemented. (Page 2, Line 62-63; reference24)

 Point 3: Result (page 4 : Line 122) it will be better if you write the aims of linearity.

 Response 3: According to your suggestion, the content of the manuscript had been supplemented. (Page 4, Line 127-128)

Within the setting concentration range, the extent of direct linear relationship between the test results and the concentration of analytes in the samples was investigated.

Point 4: Result (page 4 : Line 132) it will be better if you write the aims of stability.

 Response 4: According to your suggestion, the content of the manuscript had been supplemented and adjusted. (Page 4, Line 141-142)

To investigate whether the sample solution is stable in 24 hours being stored at room temperature.

Point 5: Result (page 5 : Line 172) According to the software requirements, it is generally requiring the similarity is greater than 0.9. In the similarity analysis, 3 batches are greater than 0.9, suggest ing that the quality of medicinal materials from different producing areas and growth conditions is quite different. Which sampel number of the 3 batches are greater then 0.9.

 Response 5: According to your suggestion, the content of the manuscript had been supplemented. (Page 5, Line 181-182)

In the similarity analysis, only 3 batches of samples were greater than 0.9, which were S1, S2 and S5, suggesting that the quality of medicinal materials from different producing areas and growth conditions is quite different.

Point 6: Result (page 6 : Line 192) Ten components were identified in the fingerprint with reference materials. You can describe what they are.

Response 6: According to your suggestion, the components had been described. (Page 6, Line 201-204)

Ten components were identified in the fingerprint with reference materials, including chlorogenic acid, apigenin-C-6,8-pentoside-hexoside, apig-enin-C-6,8-di-pentoside, luteolin, 3,4-dicaffeoylquinic acid, 3,5-dicaffeoylquinic acid, 4,5-dicaffeoylquinic acid, linarin, chrysosplenetin B, and sbsinthin.

Point 7: Result (page 7 : Line 222) The quantitative results of the 10 compounds in A. rupestris calculated by ESM and QAMS methods are shown in Tables 6 and 7, and the REs were less than 5%. does the RE requirement have to be 5%? If yes, please explain the reference of this requirement. And all the RE% results are under 5%, what it does mean?

Response 7: According to your suggestion, the content of the manuscript had been supplemented. (Page 7, Line 235-237)

The REs were less than 5%, which was the requirement of Chinese Pharmacopoeia. It was indicated that there was no significant difference in the content results obtained by QAMS and ESM.

I would like to express my sincere thanks to reviewer 1 for the recognition of the article. Thank you very much.

We thank the reviewer and remain at your disposal for any further question.

Your sincerely,

Dr. Maitinuer Maiwulanjiang

Xinjiang Key Laboratory of Plant Resources and Natural Products Chemistry,

Xinjiang Technical Institute of Physics and Chemistry,

Chinese Academy of Sciences,

Beijing South Road 40-1

Urumqi, Xinjiang, 830011, China

Phone: (+86) 0991-6631740

Fax: (+86) 0991-3838957

Email: mavlanjan@ms.xjb.ac.cn